# Intensity of Health Behaviors in People Who Practice Combat Sports and Martial Arts

**DOI:** 10.3390/ijerph16142463

**Published:** 2019-07-11

**Authors:** Katarzyna Kotarska, Leonard Nowak, Mirosława Szark-Eckardt, Maria Alicja Nowak

**Affiliations:** 1Department of Physical Culture and Health Promotion, University of Szczecin, 71-004 Szczecin, Poland; 2Kazimierz Wielki University in Bydgoszcz, 85-064 Bydgoszcz, Poland

**Keywords:** intensity of health behaviors, combat sports and martial arts, length of sport training history, socio-demographic variables, Health Behavior Inventory

## Abstract

*Background*: Health behaviors are associated with a healthy lifestyle, in which relative possibilities of choice play an important part. Athletes are a group of people who should particularly endeavor to have a health-oriented lifestyle. It is believed that combat sports (CS) and martial arts (MA) have an especially significant educational potential, connected with several desirable values which provide positive patterns of health behaviors. The aim of the work was to assess the intensity of health behaviors in athletes who practiced CS and MA in relation to the length of their training history, their age, sex, place of residence, education level, and financial situation. *Methods*: The research involved 441 men and women who practiced boxing (B), Brazilian ju-jitsu (BJJ), karate (K), mixed martial arts (MMA) and Muay Thai (MT). The average age of the subjects was 24.68 ± 8.24 years. The standardized Health Behavior Inventory (HBI) questionnaire and another questionnaire for a lifestyle survey were applied. Individual behaviors covered four areas: Correct eating habits (CEH), preventive behaviors (PB), positive mental attitude (PMA), and health practices (HP). The one-way analysis of variance (F-test) for independent groups was used (ANOVA). The effect size was calculated with Hedge’s g for Student’s t-test, and with Cramér’s V for the χ^2^ test. The value of *p* ≤ 0.05 was assumed to be statistically significant. *Results*: CS and MA athletes presented a moderate level of health behaviors. The greater intensity of health behaviors (HBI and its categories) was found among B, K and MMA athletes, and the smaller among those who practiced MT. Correct eating habits (CEH) were characteristic of subjects who practiced every day and whose length of training history was 4–8 years. Greater intensity of preventive behaviors (PB) was observed among individuals aged under-19 years, who still studied. Greater intensity of health practices (HP) was found among those who exercised every day. Influence of financial situation was observed in relations to PMA. *Conclusions*: It seems that the existing educational potential of CS and MA was not fully realized in the studied population. Determining the place of health in the system of values of CS and MA athletes may be the basis for predicting health behaviors and developing health education programs.

## 1. Introduction

Human health behaviors may be habitual (effect of socialization), reactive (response to social requirements, trends), or intentional (intentional behaviors conditioned by knowledge and health awareness) [1,2]. They are the result of relatively free personal choices and decisions concerning the avoidance of risky behaviors (use of drugs, failure to observe the principles of road safety, improper diet), choices of positive health practices (physical exercise, brushing teeth, hygiene of the environment) and preventive measures (regular checkup visits to the family doctor and dentist, self-monitoring of health, etc.) [3]. Choices of health-oriented behaviors are affected by many factors and can be quite difficult to maintain in the long term, especially if they require giving up everyday pleasures and habits [4]. In the holistic-functional model, which sees health as a process, the conscious activity of the subject who creates their health plays an important part. This approach facilitates health education, environment protection and cooperation with professional health services. The salutogenic approach also looks for health-promoting factors. According to Antonovsky, one of the factors, in addition to general immune resources, stressors, behaviors and lifestyle, is the sense of coherence, which consists of perceiving the world as understandable and controllable. A person with a strong sense of coherence can improve their functioning, through the creation of a hierarchy of certain values [5]. Antonovsky’s concept clearly emphasizes the significance of making an effort in various areas of human life, including the physical effort related to the possibility of influencing health through selection of favorable behaviors. Research into athletes’ behaviors can contribute to the promotion of their health-oriented behavior pattern.

The majority of Poles (78%) declare that they take care of their health by proper nutrition (43%), regular visits to the doctor (31%), avoiding stressful situations, not smoking cigarettes (30%, respectively), regular exercise (23%), not drinking alcoholic beverages (14%), and regular use of vitamin supplements (2%). Younger people associated maintaining health with physical activity, while older people with a better diet and medical checkups [6]. In a survey conducted by the Polish Public Opinion Research Center (CBOS) in 2016 on a representative sample of 981 Poles, it was found that 88% of the respondents did not practice any sport professionally, 58% hardly ever performed gymnastics or other exercises, and 37% did not undertake activities such as running, swimming, cycling, playing games [7]. Research has confirmed that sports activity aids health promotion and encourages the pursuit of a healthy lifestyle [8,9]. Healthy behaviors, balanced nutrition, ability to cope with stress, and behaviors harmful to health (smoking, abuse of alcoholic beverages or medicines) are particularly important for athletes [10], for whom health is a prerequisite for success in sport. Athletes are able to take every effort and engage to the maximum extent in a healthy lifestyle, because then they can handle greater optimal exercise loads and achieve sports mastery. Practicing specific disciplines, especially CS and MA, can facilitate the shaping of social attitudes and positive health behaviors [10,11,12,13].

Many authors recognize the possibilities of using CS and MA in various areas of life, such as shaping psychophysical well-being, harmony of body and mind, reducing aggression [7,11,14,15], improving the safety and quality of life of those who practice these disciplines [16]. Because martial arts are not focused on sports competition, it is a way that one can begin to follow even at a mature age. The results of the research so far have revealed that one of important factors influencing the decision to practice CS and MA was the belief in the possibility of improving health and achieving high physical fitness [17]. In Gwardyński’s study [18], it was found that judokas show a higher overall level of health-oriented behaviors than average Poles [19]. The severability of the athletes’ population and the potential of this social category make it an important research subject in the area of intensity of health behaviors.

The research so far has been characterized by limited territorial coverage and a narrow scope of research area. In this context, there is a need for in-depth studies focused on a comprehensive, multi-threaded and interdisciplinary approach to assessing the health behaviors of athletes, because individual stages of life condition their intensity, which affects both health and sports performance.

The aim of the work was to assess the intensity of health behaviors in athletes who practiced CS and MA in relation to the length of their training history, their age, sex, place of residence, education level, and financial situation.

It is supposed that athletes who practice boxing, Brazilian jiu-jitsu, karate, mixed martial arts, and Muay Thai, are characterized by varied intensity of health practices.

Greater intensity of health-oriented behaviors will be displayed by athletes with longer training histories, women, city residents, persons who are better educated, have a good financial situation and younger persons.

## 2. Material and Methods

The research embraced 441 people who practiced martial arts and combat sports in 14 clubs in the provinces of West Pomerania, Kuyavian-Pomerania and Lubuskie. The respondents practiced boxing (B), Brazilian jiu-jitsu (BJJ), karate (K), mixed martial arts (MMA), and Muay Thai, i.e., Thai boxing (MT). The respondents were at various sporting levels: Competitors who had not had significant sports achievements (375 individuals), and 66 competitors with significant sports achievements, at the level of at least the Polish Championships. The average age of the subjects was 24.68 ± 8.24 years. Men accounted for 52.6% of the respondents. The majority were unmarried (72.5%). Among the subjects, 52.8% still studied, but 14.5% of those who were in schools also worked. 40.2% were employed (25.2% did mental and 15% physical work), 3.6% ran their own businesses. Subjects who were between jobs and those who chose not to work constituted 3.4%. The standardized health behavior inventory (HBI) was used to assess health behaviors and their intensity [19]. The HBI internal consistency, based on Cronbach’s alpha, was 0.85 for the entire HBI, and 0.65, 0.61, 0.60, and 0.64 for its four subscales (CEH, PB, PMA, HP. The reliability assessed by the test–retest method (after six weeks) was 0.88. The use of the Cronbach Alpha formula was preceded by a factor analysis [19]. In our research, Cronbach’s alpha was as follows: HBI (point score): 0.868, CEH: 0.778, PB: 0.641, PMA: 0.712, and HP: 0.605.

Due to the underestimation of reliability by Cronbach’s alpha [20], we used McDonald’s ω [21] which was higher in each category: HBI (point score): 0.875, CEH: 0.787, PB: 0.660, PMA: 0.723, and HB: 0.633. Individual behaviors covered four areas: Correct eating habits (CEH) mainly concern the type of food consumed, such as vegetables, fruit, animal fats, sugars, salt, whole wheat bread; preventive behaviors (PB) include compliance with medical recommendations, avoiding colds, knowledge of emergency numbers, regular medical examinations; positive mental attitude (PMA) is about avoiding too-strong emotions, depressing situations, stress and tension; and health practices (HP) include daily habits relating to rest, sleep, recreation, physical activity, maintaining proper body weight, using stimulants, and smoking. Each behavior was rated on a five-point scale. On the basis of the frequency of indicated health behaviors, the general intensity of the particular categories of behaviors conducive to health was determined, as well as the level of intensity of individual categories [19]. The general intensity of these behaviors was the sum of points from the individual categories, within the range from 24 to 120 points. A higher result denotes a greater intensity of declared behaviors. In addition, an original survey technique was used to survey the lifestyles of the athletes. This study used data on age, sex, place of residence, education, type of work and financial situation as well as information on sporting disciplines, sporting experience, and weekly exercise time. The results obtained with the help of HBI and its categories were referred to these variables.

Written informed consent was obtained from each subject included in the study. The study protocol was approved by the appropriate Ethics Committee of Kazimierz Wielki University No. KEBN 7/2018 and conformed to the ethical guidelines of the 1975 Declaration of Helsinki.

The one-way analysis of variance (F-test), confirmatory factor analysis for independent groups was used (ANOVA), and Student’s t-test. In qualitative analyses, the trait frequency and the independence chi-square test were used. The effect size was calculated Hedges’g (g) for Student’s t-test, and Cramér’s V for the χ^2^ test. The value of *p* ≤ 0.05 was assumed to be statistically significant. Statistical calculations were made with the Statistica for Window 12 software (StatSoft Sp. z o.o., Crakow, Poland), and Microsoft Office Excel 2007 (Microsoft Sp. z o.o., Warsaw, Poland) and JASP 0.10.0.0 (https://jasp-stats.org).

## 3. Results

Karate athletes (K) were the youngest (more than half of them aged 19 and below) (*p* = 0.0000 for the χ^2^ test), boxers (B) were more often aged 20–23 (Table 1). Those who practiced BJJ, MMA, and MT were aged 24–28, but there was also a similar number of the last mentioned in the oldest group—over 28 years old. While the majority of the respondents lived in cities, karate athletes more often than others came from the countryside (20.4%) (*p* = 0.0086 for the χ^2^ test). The respondents had a post-secondary (46%) and secondary education (35.1%). The majority of BJJ and MMA athletes had a post-secondary education (*p* = 0.0000 for the χ^2^ test). Over 1/3 of karate athletes continued their education in secondary schools. The subjects had a good (47.1%) and very good financial situation (35.8%). Karate athletes less often assessed their financial situation as fair (*p* = 0.0070 for the χ^2^ test). Half of combat sports athletes practiced 3–4 times a week. The ones who practiced every day where mainly boxers (56.5%) (*p* = 0.0000 for the χ^2^ test). For 46.7% of the respondents, the weekly exercise time was less than 240 min. The group of persons training for between 241 and 360 min per week included BJJ, MMA and MT athletes. The ones who practiced for more than 481 min were mainly MMA and B practitioners (29% and 26.1%, respectively) (*p* = 0.0000 for the χ^2^ test). The respondents were characterized by varied lengths of practicing sport. BJJ, MT, and K athletes (38.3% in total) had practiced for less than 2 years. 37.7% of boxers had practiced for 5 to 8 years. Each group included approximately 15% of athletes who had practiced combat sports for more than 8 years (*p* = 0.0000 for the χ^2^ test). The results show a medium effect size for the respondents’ education and weekly exercise frequency, and a small effect size for the other variables analyzed.

The applied analysis of variance confirmed the differences between the intensity of the health behaviors of different athletes (Table 2). In further analysis we will show the effect size with Hedge’s g for Student’s t-test when the effect size is at the medium level (g = 0.5) or close to medium (g = 0.4). MT (Thai boxing) athletes achieved the lowest arithmetic means in all Health Behavior Inventory (HBI) categories, while the highest were obtained by boxers, with the exception of preventive behaviors (PB) (better results were achieved by karate athletes) and positive mental attitude (PMA) (better results were achieved by MMA practitioners). HBI (point score) for MT were statistically lower in comparison with K (*p* = 0.0013; g = −0.4), and B (*p* = 0.0051; g = −0.4). The effect size between the intensity of health behaviors (HBI point score) of boxers, karate and MT practitioners was found close to medium (Hedge’s g = −0.4). As far as correct eating habits (CEH) are concerned, lower levels were presented by MT athletes compared to B (*p* = 0.0023; g = −0.5) and MMA (*p* = 0.0217; g = 0.5) practitioners as well as BJJ athletes compared to B (*p* = 0.0200; g = −0.4) and MMA athletes (g = −0.4). For preventive behaviors (PB), BJJ athletes achieved significantly lower results in comparison with K (*p* = 0.0036; g = −0.4) and B athletes, similarly to MT athletes, also in comparison with K (*p* = 0.0025; g = −0.4) and B practitioners. In the positive mental attitude (PMA) category, lower scores were achieved by MT athletes compared to K practitioners. Lower rank means for health practices (HP) were achieved by MT athletes compared to BJJ, K and B (*p* = 0.0089; g = −0.4) athletes.

The respondents’ health behaviors in relation to the length of their training history are presented in Table 3. The highest arithmetic means for HBI (point score) were achieved by athletes with 4–8 years of training history, who presented a higher level of health behaviors in comparison with athletes who had been practicing longer—more than 8 years (*p* = 0.0225; g = 0.4). Also, the athletes with less than 2 years and between 2–4 years of training history had significantly lower results in comparison with those who had been training for 4–8 years. Correct eating habits (CEH) were presented more often by persons who had been practicing for 4–8 years in comparison with those who had for ≤2 years (*p* = 0.0031; g = –0.4) and for more than 8 years (*p* = 0.0182; g = 0.4). As for preventive behaviors (PB), better results were obtained by persons who had been practicing for 4–8 years than by those who had for 2–4 years and ≤2 years. Positive mental attitude (PMA) was characteristic of those who had been training for 4–8 years rather than those with a shorter training history (2–4; and ≤2 years;), or those who had been training for more than 8 years. In the health practices (HP) category, CS and MA athletes who had been training for 4–8 years achieved better results than those with a training history of ≤2 years. Better results concerning HBI (point score) and CEH were achieved by athletes with training experience 4–8 years compared to those who had practiced for over 8 and less than 2 years (effect size close to medium). However, the differences between PB, PMA and HP and the length of training history of the respondents show a small effect size.

The health behaviors of CS and MA athletes in relation to weekly training frequency is presented in Table 4**.** The overall HBI point score rank means were higher for athletes who practiced every day than for those who practiced 3–4 times a week and twice a week (*p* = 0.0194; *p* = 0.0054; respectively; g = 0.4). Athletes who trained every day presented more correct eating habits than those who trained 3–4 times a week and 2 times a week (*p* = 0.0011; g = 0.4) and achieved higher results in preventive behaviors (PB) than those who trained 3–4 times a week (*p* = 0.0129). Within the health practices (HP) category, athletes who trained every day (*p* = 0.0037; g = 0.4) achieved higher results in comparison with those who practiced 2 times a week. Greater intensity of health behaviors among athletes who practiced every day in comparison with those who practiced 2 times a week is confirmed by close to medium effect size with respect to HBI point score, CEH and HP.

It was also found that age had an influence on the intensity of health behaviors (Table 5). Persons aged under-19 were characterized by greater intensity of health behaviors (HBI point score) in comparison with subjects aged 24–28 years (*p* = 0.0032; g = 0.4) and over 28. As for preventive behaviors (PB), the youngest athletes achieved better results than persons aged 24–28 years (*p* = 0.0002; g = 0.4) and over 28 years. The effect size was close to medium.

Differences were observed between the arithmetic means for persons with a pre-secondary education and those with a secondary or post-secondary education (HBI point score: *p* = 0.0108, g = 0.4; *p* = 0.0037, g = 0.4; and PB: *p* = 0.0042, g = 0.4) (Table 6). The effect size was close to medium. Individuals who continued their education and had a pre-secondary education achieved higher intensity of health behaviors than those with a secondary or post-secondary education, also with regard to PMA and HP. In this case, the effect size was small.

Persons who assessed their financial situation as very good and good were characterized by greater intensity of health behaviors in comparison with those who had a sufficient financial situation with regard to HBI point score (*p* = 0.0011; g = 0.5). Greater PMA intensity was observed in persons having the best financial situation compared to those with a good financial situation (*p* = 0.0001; g = 0.5). Individuals who had a sufficient financial situation also had lower PMA scores than those who had a good financial situation. A medium effect size with respect to HBI point score and PMA was found in relation to the analyzed variables (greater intensity of health behaviors in individuals in a very good situation in comparison to those who were worse off) (Table 7).

Differences between the intensity of correct eating habits (CEH) of women and men were confirmed (F-test = 5.6779; *p* = 0.0176 *). The arithmetic mean for men amounted to 21.02, and for women 19.98. Women achieved worse results in this category (small effect size). Differences between the intensity of HBI and its categories and CS and MA athletes’ place of residence were not confirmed.

The interaction of factors produces an additional effect (Table 8). Sex, age, education, material situation, frequency and history of training had an impact on the intensity of health behaviors of CS and MA athletes. The two-factor interaction between sport discipline and sex was modified by age factor (CEH, PB, PMA) and the interaction between sport and age was modified by sex (PMA). The two-factor interaction between sports discipline and frequency of exercises was modified by sporting experience (CEH, HP) and sporting experience and sex (CEH, PB, HP). 

Three-factor interactions: between sport discipline, age and education was modified by sex (CEH); between sport, age and sex was modified by financial situation (CEH, PB); between sport, age and financial situation was modified by sex (HP); interaction between sport, frequency and seniority is was modified by sex (CEH, PB, PMA). We only found an interaction between sport discipline, weeky frequency of training and sporting experience (PMA).

Sex was the factor which most often modified the analyzed interactions (two- and three-factor interactions).

## 4. Discussion

Combat sports and martial arts seem to have a special educational value, as they often focus not only on tackling an opponent, but also on shaping the personality and caring about one’s own body. CS and MA seem to have a special educational value because they focus not only on competing against an opponent, but also on shaping personality, taking care of one’s own body, safety and, if necessary, self-defence [22,23]. People who practice sports usually pay attention to proper nutrition, avoiding stimulants, regular rest and selection of leisure activities [24]. Our research has confirmed the hypothesis that health behaviors depend on the CS and MA practiced. Athletes who practiced MT (Thai boxing) achieved the lowest intensity of health behaviors in all HBI categories, similarly to BJJ practitioners (except for PMA). The greatest intensity of health behaviors in the HBI and its categories was characteristic of boxers and karate athletes. MMA athletes had better consolidated eating habits compared to those who practiced MT. The detailed analysis of all categories of health behaviors in the sample group of athletes showed that the highest arithmetic mean was obtained for PMA. This indicator plays a significant motivational role and influences the results of the collected experiences and activities of athletes [25]. This is confirmed by our results: The examined athletes achieved high means in positive mental attitude (PMA). It should be assumed that they are ready to engage in health-oriented behaviors.

MT and BJJ are among the most brutal sports and do not cultivate Eastern traditions or ceremonies. MT is considered the most aggressive, forceful and effective martial art in the world, attracting young people with spectacular and violent knockouts. An advantage of MT training is the increase in the athletes’ self-confidence. BJJ is considered the most effective art of hand-to-hand combat in a real fight. The goal of the bout is to take control of the opponent by giving him controlled pain. BJJ owes its current popularity above all to the great effectiveness of the athletes of this discipline in MMA fights. Most of the training is conducted in the form of fights. It is appreciated by young people, who constantly compete and confirm their skills [12,26]

Greater intensity of health behaviors in karate probably stems from the fact that karate, more than other CS and MA, focuses not only on harmonious physical development, but also aims to enhance health [27]. Athletes with a more instrumental approach recognize other values, create various hierarchies and choose different behaviors from those that follow a certain philosophy or ideology. In the light of research, karate athletes declared a change in the approach to everyday problems and a change in lifestyle as a result of practicing this martial art [17].

Among the examined karate athletes, there was the largest number of those who still studied, practiced 3-4 times a week, and spent about 6 h training. Almost half of them had practiced this sport for about 2 years. It is possible that this group of respondents, subjected to various educational influences at school and in training, chose more correct health behaviors. Research highlights the need to combine sports training with personality development in order to prepare athletes for social life. Increasing the level of health education in the CS and MA training process is expected by athletes and appreciated by coaches [28]. Research undertaken by Sterkowicz [29] showed that the karate value system was dependent on age, training experience, the role played in the club, gender and the level of advancement. Out of 56 values, the most popular were perseverance, attachment to tradition, good reputation, contemplation, society-oriented attitude, respect for authorities, individuality, simplicity, truthfulness, happy companionship, empathy, appearance, bravery and steadfastness. The largest statistically significant differences were related to age; impact of age was demonstrated in 9 out of 15 factors. Training experience had a positive correlation with attachment to tradition, honesty and simplicity. Research showed that karate practitioners are calm people characterized by low nervous tension, low anxiety, great self-confidence and a sense of security [30]. Research conducted by students of the Medical University of Lublin showed that people from rural areas display more favourable health behaviors related to correct eating habits and preventive behaviors [31]. These results were not confirmed in our research. There were no differences in the intensity of health behaviors of athletes depending on their place of residence.

Boxers had similar results to karate athletes with regard to HBI (point score), and preventive behaviors (PB). The majority of the boxers graduated from a secondary school, had a good financial situation, and practiced every day. About 40% had 4–8 years of training experience.

Boxing has rich sporting traditions, as well as improved rules and refereeing methods. Sound psychological and physical preparation of boxers and proper medical care have a significant impact on the selection of this discipline and the choices of health behaviors of its practitioners [17] Our study confirmed the assumption that a higher level of health behaviors would be presented by athletes with a longer training history. Athletes with 4–8 years of experience in practicing sports, achieved higher results in HBI and its categories in comparison with those who had been training for more than 8 years. MMA athletes showed similar behaviors to boxers and karate athletes (HBI and categories). Nearly half of them had practiced this sport for more than 4 years. Among MMA and MT athletes, women constituted a minority (35.5%; 40% respectively). MT and BJJ athletes had less training experience.

Correct eating habits (CEH) were characteristic of those who practiced every day, and had 4–8 years of training experience. Physically active people are more optimistic compared to people who do not practice sports. At the same time, optimism correlates with health-oriented practices, such as every day habits concerning sleep and rest, engaging in physical activity and correct eating habits [32]. As far as determinants of health behaviors are concerned, the influence of age, sex, family situation, education, occupation and financial situation are most often emphasized [33]. These factors differentiate attitudes towards health and illness, physical activity, and free time. A higher economic status facilitates the implementation of health-oriented behaviors through easier access to medicines, various forms of sporting activity, or having a healthy diet [11], but with insufficient health awareness and lack of habits, it does not guarantee that the right choices will be made. The relationship between the intensity of health behaviors and these variables was confirmed in our research. The youngest individuals, aged under-19, were characterized by greater intensity of health behaviors in comparison with those aged 24–28 and over 28. As regards PB (preventive behaviors), these differences were the greatest (confirmed by a medium effect size)**.** Our observations concerning the positive influence of young age on health-oriented attitudes are confirmed by the results of research of junior sumo athletes [34]. At the same time, it was observed that the intensity of health behaviors (also in individual categories) decreased with age [12]. Individuals who still studied, with a pre-secondary education, achieved higher intensity of health behaviors compared to those with a higher education, also in the area of PMA and HP. Undertaking physical activity depends to a large extent on the social environment that shapes and preserves the behavior patterns of an adolescent. The strength of its impact varies and depends on the age and the place and position of the individual in a particular social group. All young people participate in the education process, where physical activity is obligatory. They do not make independent choices. This group includes mainly school and university students, who do not take up paid work due to their studies [35]. Differences between the intensity of correct eating habits of women and men were confirmed. Women achived worse results in this category. There are other results available in literature that show higher intake of fiber-rich foods, poultry and fish, and fruit and vegetables by women. In addition, women focus on preventive behaviors more often than men [36]. This issue requires further research and clarification regarding the impact of gender on health behaviors of women who practice CS and MA. However, the weekly training frequency had a positive effect on the intensity of health behaviors. Respondents who practiced CS and MA every day, presented more correct behaviors with respect to HBI and each category than those who practiced less frequently (3–4 times a week, 2 times a week).

CS and MA athletes presented moderate level of intensity of health behaviors. Our results, however, are consistent with some studies which show a high level of unhealthy behavior of athletes and participants of sports courses, despite the fact that these people should be potentially well prepared to play the role of promoters of a healthy lifestyle [36]; the same concerns students of medical universities [31] who should be well prepared to play the role of promoters of a healthy lifestyle.

### Limitation

Future research should determine the relationships between the attitude of subjects toward professional work (mental, physical, unemployed) and learning and practicing CS and MA—in the context of sports training objectives. It would also be important to investigate the motives for CS and MA practicing by women and inhabitants of rural areas, which would facilitate a broader assessment of health behaviors and their determinants. In addition, determining the place of health in the system of values of CS and MA athletes may be the basis for predicting health behaviors and developing health education programs.

## 5. Conclusions

The greater intensity of health behaviors (HBI and its categories) was found among B, K and MMA athletes, and the smaller among those who practiced MT. Correct eating habits (CEH) were characteristic of subjects who practiced every day and whose length of training history was 4–8 years. Greater intensity of preventive behaviors (PB) was observed among individuals aged under-19 years, who still studied, and of health practices (HP) among those who exercised every day. Influence of financial situation was observed in relations to PMA. CS and MA athletes presented a moderate level of health behaviors, despite the differences between K, B, MMA, MT, and BJJ practitioners.

Significantly higher scores in the overall HBI and all its categories were achieved by athletes practicing every day. Determinants of intensity of health behaviors of CS and MA athletes require further research, especially in the context of the self-assessment of their health.

## Figures and Tables

**Table 1 ijerph-16-02463-t001:** Characteristics of people who practice combat sports (CS) and MA (independence χ^2^ test, and Cramér’s V).

Variables	CS and MS (%)					Total (441)		*p* for χ^2^	Cramér’s V
	BJJ (*n* = 90)	MMA (*n* = 31)	MT (*n* = 104)	K (*n* = 147)	B (*n* = 69)	*N*	%		
Age:								0.0000	0.2
<19	10	19.3	20.2	51.8	24.6	129	29.3		
20–23	20	25.8	19.2	12.9	34.8	89	20.2		
24–28	42.2	32.3	29.8	12.2	23.2	113	25.6		
>28	27.8	22.6	30.8	23.1	17.4	110	24.9		
Sex:								n.s.	0.1
- men	47.7	64.5	58.7	49	52.2	232	52.6		
- women	52.2	35.5	41.3	51	47.8	209	47.4		
Place of residence:								0.0086	0.2
- city	90	87.1	94.1	79.6	91.3	383	87.4		
- village	10	12.9	5.9	20.4	8.7	55	12.6		
Education:								0.0000	0.3
-pre-secondary	8.9	3.2	11	37	12.1	82	18.9		
- secondary	28.9	38.7	45	21.9	56.1	152	35.1		
- post-secondary	62.2	58.1	44	41.1	31.8	199	46		
Financial situation:								0.0070	0.2
- very good	28.9	20	39.2	44.9	27.5	157	35.8		
- good	50	56.7	38.2	45.6	55.1	206	47		
- fair	21.1	23.3	22.6	9.5	17.4	75	17.1		
Weekly exercise frequency:								0.0000	0.3
- every day	21.1	29.1	17.3	17	56.5	110	24.9		
- 3–4 times a week	51.1	64.5	59.6	46.9	36.2	222	50.3		
- 1–2 times a week	27.8	6.4	23.1	36.1	7.3	109	24.8		
Length of practicing sport:								0.0755	0.1
≤2 years	37.8	22.6	38.5	47.6	26.1	169	38.3		
>2 <4 years	16.7	29	25	19	21.7	93	21.1		
>4 <8 years	28.8	32.3	22.1	18.4	37.7	112	25.4		
≥8 years	16.7	16.1	14.4	15	14.5	67	15.2		

BJJ: Brazilian ju-jitsu; MMA: mixed martial arts; MT: Thai boxing; K: karate; B: boxers.

**Table 2 ijerph-16-02463-t002:** Health behaviors of people who practice CS and MS (F-test, Student’s t-test, Hedge’s g).

HBI and Its Categories	Sports	MMA	MT	K	B	MMA	MT	K	B	Means
		Student’s t-Test				Hedge’s g				
HBI (point score) F = 3.7506 *p* = 0.0052 *	BJJ	0.4009	0.1720	0.0976	0.0795	−0.2	0.2	−0.2	−0.3	80.86
	MMA		0.0983	0.8451	0.5808		0.3	0.0	−0.1	83.10
	MT			0.0013 *	0.0051 *			−0.4	−0.4	77.86
	K				0.5128				−0.1	83.61
	B									84.83
CEH F = 3.9113 *p* = 0.0039 *	BJJ	0.0707	0.3542	0.1369	0.0200 *	−0.4	0.1	−0.2	−0.4	20.00
	MMA		0.0217 *	0.3319	0.9025		0.5	0.2	0.0	21.63
	MT			0.0108 *	0.0023 *			−0.3	−0.5	19.36
	K				0.1600				−0.2	20.86
	B									21.77
PB F = 3.5812 *p* = 0.0069 *	BJJ	0.3280	0.8463	0.0036 *	0.0394 *	−0.2	0.0	−0.4	−0.3	18.86
	MMA		0.3220	0.3389	0.5551		0.2	−0.2	−0.1	19.67
	MT			0.0025*	0.0387 *			−0.4	−0.3	18.72
	K				0.6770				0.1	20.57
	B									20.30
PMA F = 1.3996 *p* = 0.2332	BJJ	0.5560	0.2181	0.5178	0.5187	−0.1	0.2	−0.1	−0.1	21.64
	MMA		0.1792	0.8245	0.9525		0.3	0.0	0.0	22.10
	MT			0.0382 *	0.0943			−0.3	−0.3	20.88
	K				0.8520				0.0	21.97
	B									22.07
HP F = 2.7944 *p* = 0.0259 *	BJJ	0.4424	0.0219 *	0.7832	0.6038	0.2	0.3	0.0	−0.1	20.36
	MMA		0.3957	0.4797	0.2483		0.2	-0.1	−0.3	19.70
	MT			0.0116 *	0.0089 *			-0.3	−0.4	18.89
	K				0.3819				−0.1	20.22
	B									20.68

* Statistically significant for *p* ≤ 0.05.

**Table 3 ijerph-16-02463-t003:** Health behaviors of respondents with varied length of training history (F-test, Student’s t-test, Hedge’s g).

HBI and Its Categories	Length of Training History	Student’s t-Test			Hedge’s g			Means
		>2 <4 Years	>4 <8 Years	≥8 Years	>2 <4 Years	>4 <8 Years	≥8 Years	
HBI (point score) F = 3.2466 *p* = 0.0219 *	≤2 years	0.6112	0.0046 *	0.8566	−0.1	−0.3	0.0	80.32
	2–4 years		0.0269 *	0.7850		−0.3	0.0	81.27
	4–8 years			0.0225 *			0.4	85.33
	≥8 years							80.70
CEH F = 2.7622 *p* = 0.0415 *	≤2 years	0.2303	0.0031 *	0.9765	−0.2	−0.4	0.0	19.94
	2–4 years		0.1477	0.3468		−0.2	0.2	20.67
	4–8 years			0.0182 *			0.4	21.59
	≥8 years							19.96
PB F = 1.6636 *p* = 0.1738	≤2 years	0.7474	0.0491 *	0.7544	0.0	−0.2	0.0	19.47
	2–4 years		0.0340 *	0.9837		−0.3	0.0	19.27
	4–8 years			0.0436 *			0.3	20.59
	≥8 years							19.25
PMA F = 1.4630 *p* = 0.2237	≤2 years	0.8025	0.0261 *	0.9766	0.0	−0.3	0.0	21.41
	2–4 years		0.0205 *	0.8454		−0.3	0.0	21.27
	4–8 years			0.0492 *			0.3	22.56
	≥8 years							21.39
HP F = 1.8941 *p* = 0.1298	≤2 years	0.2312	0.0234 *	0.2619	−0.2	−0.3	−0.2	19.45
	2–4 years		0.3424	0.9481		−0.1	0.0	20.06
	4–8 years			0.4350			0.1	20.60
	≥8 years							20.10

* Statistically significant for *p* ≤ 0.05.

**Table 4 ijerph-16-02463-t004:** Health behaviors of people who practice CS and MA, in relation to weekly training frequency (F-test, Student’s t-test, Hedge’s g).

HBI and Its Categories	Frequency of Exercises a Week	Student’s t-Test		Hedge’s g		Means
		3–4 Times a Week	2 Times a Week	3–4 Times a Week	2 Times a Week	
HBI (point score) F = 4.2417 *p* = 0.0150 *	everyday	0.0194 *	0.0054 *	0.3	0.4	85.03
	3–4 times a week		0.3829		0.1	81.27
	2 times a week					79.83
CEH F = 5.2988 *p* = 0.0053 *	everyday	0.0375 *	0.0011 *	0.2	0.4	21.39
	3–4 times a week		0.1074		0.2	20.40
	2 times a week					19.60
PB F = 3.9389 *p* = 0.0200 *	everyday	0.0047 *	0.2939	0.3	0.1	20.28
	3–4 times a week		0.1593		−0.2	19.01
	2 times a week					19.72
HP F = 4.0680 *p* = 0.0178 *	everyday	0.3451	0.0037 *	0.1	0.4	20.56
	3–4 times a week		0.0324 *		0.3	20.12
	2 times a week					19.07

* Statistically significant for *p* ≤ 0.05.

**Table 5 ijerph-16-02463-t005:** Health behaviors of people who practice CS and MA, in relation to age (F-test, Student’s t-test, Hedge’s g).

HBI and Its Categories	Wiek/Age	Student’s t-Test			Hedge’s g			Means
		20–23 Years	24–28 Years	>28 Years	20–23 Years	24–28 Years	>28 Years	
HBI (point score) F = 3.6207 *p* = 0.0132 *	<19	0.4148	0.0032 *	0.0076 *	0.1	0.4	0.3	84.58
	20–23 years		0.0876	0.1315		0.2	0.2	83.12
	24–28 years			0.8902			0.0	79.63
	>28 years							79.90
PB F = 4.7963 *p* = 0.0026 *	<19	0.0905	0.0002 *	0.0043 *	0.2	0.4	0.3	20.56
	20–23 years		0.0964	0.2729		0.2	0.1	19.67
	24–28 years			0.6878			0.0	18.75
	>28 years							18.98

* Statistically significant for *p* ≤ 0.05.

**Table 6 ijerph-16-02463-t006:** Health behaviors of people who practice CS and MA, in relation to the level of education (F-test, Student’s t-test, Hedge’s g).

HBI and Its Categories	Education	Student’s t-Test		Hedge’s g		Means
		Secondary	Post-Secondary	Secondary	Post-Secondary	
HBI (point score) F = 4.4478 *p* = 0.0122 *	pre-secondary	0.0108 *	0.0037 *	0.4	0.4	85.89
	secondary		0.5417		0.1	81.47
	post-secondary					80.50
PB F = 4.5799 *p* = 0.0107 *	pre-secondary	0.0042 *	0.0051 *	0.4	0.3	20.84
	secondary		0.9293		0.0	19.28
	post-secondary					19.24
PMA F = 3.4209 *p* = 0.0334 *	pre-secondary	0.0117 *	0.0189 *	0.3	0.3	22.60
	secondary		0.8800		0.0	21.34
	post-secondary					21.40
HP F = 3.3559 *p* = 0.0358 *	pre-secondary	0.0308 *	0.0113 *	0.3	0.3	20.99
	secondary		0.6539		0.0	19.84
	post-secondary					19.64

* Statistically significant for *p* ≤ 0.05.

**Table 7 ijerph-16-02463-t007:** Health behaviors of people who practice CS and MA, in relation to the self-evaluation of their financial situation (F-test, Student’s t-test, Hedge’s g).

HBI and Its Categories	Specification	Student’s t-Test		Hedge’s g		Means
		Good	Sufficient	Good	Sufficient	
HBI (point score) F = 5.3283 *p* = 0.0052 *	very good	0.1176	0.0011 *	0.2	0.5	83.99
	good		0.0329 *		0.3	81.68
	sufficient					77.65
PNP F = 7.9634 *p* = 0.0004 *	very good	0.0132	0.0001 *	0.2	0.5	22.41
	good		0.0379 *		0.2	21.43
	sufficient					20.42

* Statistically significant for *p* ≤ 0.05.

**Table 8 ijerph-16-02463-t008:** Statistically significant interactions between the selected health behavior categories (confirmatory factor analysis).

Combinations of Factors	Interactions	Categories of Health Behaviors
CEH	PB	PMA	HP
F	*p*	F	*p*	F	*p*	F	*p*
sport*age*sex	sport*sex	4.30	0.0020	2.73	0.0289	3.31	0.0109		
sport*age					1.83	0.0411		
sport*age*sex* education	sport*age*education	2.52	0.0409						
sport*age*sex* financial situation	sport*age*sex	7.13	0.0079	5.24	0.0227				
sport*age*financial situation							2.11	0.0233
sport*weekly frequency*sporting experience	sport*weekly frequency* sporting experience					1.66	0.0368		
sport*weekly frequency	3.77	0.0024					3.06	0.0101
sport*sporting experience*weekly frequency*sex	sport*weekly frequency	8.22	0.0044	5.03	0.0256			4.75	0.0300
sport*sporting experience*weekly frequency	1.96	0.0314	1.87	0.0425	2.75	0.0020

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
