# Peer review of "Intensity of Health Behaviors in People Who Practice Combat Sports and Martial Arts"

_ijerph, 2019, doi:10.3390/ijerph16142463_

Round 1
Reviewer 1 Report
Using a sten score is unnecessary.(to show differences between groups) - raw results show this much more accurately.
In my opinion the title of this article it doesn't fully reflect the aim of this study and used methodology. Authors focused primarily on showing the level of health behaviours, not on searching for determinants.
In summary authors didn't characterize the research tool, but unnecessarily focused on describing of statistical methods.
Author Response
Dear Editor,
Thank you very much for proposed corrections. We have substantially revised the manuscript according to your suggestions. Below are the detailed answers to issues which were raised.
We hope that above explanation and corrections done have met your expectations.
Best regards
Katarzyna Kotarska

Reviewer 2 Report
The paper aims to assess the intensity of health behaviors and their socio-demographic determinants among athletes who practice CS and MA.I would like to commend the authors on the recruitment of large sample size and undertaking of complex analyses. Unfortunately, the paper is difficult to follow at times due to issues with sentence structure and language and is highly repetitive in places.
Introduction
1. This is a very difficult paper to follow. At various points in the Introduction. In relation to the contribution of the study to the literature, I did not get a sense from the article that the findings revealed anything other than what we already know.
2. The introduction of the paper was very descriptive, it did not situate the current study in literature or highlight what the gap in the literature is that this study is trying to address
3. What is the theoretical framework that supports the present study?
4. Why did you believe on these hypotheses? The hypotheses of the present study must be better described, in terms of why you believe?
Methodology
Regarding this sub-heading, it is very difficult to follow. Authors reported that the following sentence: “The diagnostic survey method was used, based on a standardized health behavior inventory(HBI) consisting of 24 questions, to assess health behaviors and their intensity.”, So, this measure was developed by who? Additionally, the composite reliability of each factor of the questionnaire must be reported. Besides that the overall adjustment values of the questionnaire in the present sample must be included.
Lines 135 – 142 I understand that the variables are asymmetric. However, considering your sample size, I suggest a parametric technique, since are stronger than non-parametric techniques.
Line 139: Why authors used this kind of measure to determine effect size?
Results
The results section is well described. However, considering my previous comment, probably the results must change in terms of values and tests.
Discussion
I will not provide detailed comments on the discussion as I believe it is likely to change quite a bit if you follow my previous recommendations. But I will provide some general comments.
I also suggest that you order the discussion according to the study aims and use sub-headings for each part of the discussion.
Overall, the discussion is very descriptive and any statements about the contribution and conclusions of the study are not new. What is the contribution to the literature, what is interesting about your results? The issues with the introduction about the lack of appropriate literature on the motivational sequence are replicated in the discussion.
Author Response

(The authors gave the same response as above.)

Reviewer 3 Report
The study gives an extensive description of (relative) Health Behaviour of athletes involved in combat sports/martial arts.
The relevance of health behaviour and healthy lifestyle is undisputed, and I see the relevance of studying the relationship between health behaviour and sport participation, age, gender, training frequency and socio-demographic variables. But to be honest I find it hard to find an argumentation to focus on CS and MA and even more why there is a focus on the differences between the different types of CS and MA. Is it possible to provide more argumentation in the Introduction? Are there indications in other sports that there are strong differences between different types of sport with respect to health behaviour?
In my opinion a descriptive approach to the data would lead to more insight in the 'determinants' of health behaviour than the extensive testing (Rank Means, U statistics, rg). I hardly find any references in the text to rg, what is the message for the reader based on the Glass Rank Biserial Correlation???
A much simpeler presentation of the data would give more insight. Why is only the Mean HBI point score calculated for the group as a whole. This value 81.85 can be related to other data as the authors show in the discussion, the rank means are less informative. For me it would be informative to see this arithmetic mean of the HBI point score for Men and for Women etc. (Even without statistical testing).
Some of the references are in the polish language. Please add only references with at least a title and summary in English.
Author Response

(The authors gave the same response as above.)

Round 2
Reviewer 2 Report
I highly appreciate the authors’ sincere responses to my previous comments. In this revision, it was much easier to understand the statistical methods and the explanation of the contents. I think the paper is in better shape; however, the following minor comments need to be checked. I would highly appreciate if the following point is addressed.
In the first revision I did the following comment: "Regarding this sub-heading, it is very difficult to follow. Authors reported that the following sentence: “The
diagnostic survey method was used, based on a standardized health behavior inventory(HBI) consisting of 24 questions, to assess health behaviors and their intensity.”, So, this measure was developed by who? Additionally, the composite reliability of each factor of the questionnaire must be reported. Besides that, the overall adjustment values of the questionnaire in the present sample must be included. "
However, the authors do not address my concern. In line with that, I suggest that authors provided composite reliability of each of sub.scale and the entire HBI via Raykov (1997) formula and not Cronbach Alpha. Additionally, the authors must provide the global adjustment of the questionnaire via a CFA (confirmatory factor analysis).
Author Response
Dear Editor,
Thank you very much for proposed corrections. We have substantially revised the manuscript according to your suggestions. Below are the detailed answers and corrections. We hope that thanks to them the paper better explains the relations between the variables.
Comments and Suggestions for Authors
1. I highly appreciate the authors’ sincere responses to my previous comments. In this revision, it was much easier to understand the statistical methods and the explanation of the contents. I think the paper is in better shape; however, the following minor comments need to be checked. I would highly appreciate if the following point is addressed.
2. In the first revision I did the following comment: "Regarding this sub-heading, it is very difficult to follow. Authors reported that the following sentence: “The
diagnostic survey method was used, based on a standardized health behavior inventory(HBI) consisting of 24 questions, to assess health behaviors and their intensity.”, So, this measure was developed by who? Additionally, the composite reliability of each factor of the questionnaire must be reported. Besides that, the overall adjustment values of the questionnaire in the present sample must be included. "
3. However, the authors do not address my concern. In line with that, I suggest that authors provided composite reliability of each of sub.scale and the entire HBI via Raykov (1997) formula and not Cronbach Alpha. Additionally, the authors must provide the global adjustment of the questionnaire via a CFA (confirmatory factor analysis).
Answers to Reviewer:
The standardized health behavior inventory (HBI) is used to assess health behaviors and their intensity [Juczyński 2012]. The HBI internal consistency, based on Cronbach’s alpha, is 0.85 for the entire HBI, and 0.65, 0.61, 0.60 and 0.64 for its four subscales (CEH, PB, PMA, HP). The reliability assessed by the test-retest method (after six weeks) was 0.88 The use of the Cronbach Alpha formula was preceded by a factor analysis [Juczyński 2012]. In our research, Cronbach’s alpha was as follows: HBI (point score) – 0.868, CEH – 0.778, PB – 0.641, PMA – 0.712, and HB – 0.605.
Due to the underestimation of reliability by Cronbach's alpha [Raykov 1997], we used McDonald’s ω [Ciżkowicz 2018], which was higher in each category: HBI (point score) – 0.875, CEH – 0.787, PB – 0.660, PMA – 0.723, and HB – 0.633.
Individual behaviors covered four areas. Correct eating habits (CEH) mainly concern the type of food consumed, such as vegetables, fruit, animal fats, sugars, salt, whole wheat bread. Preventive behaviors (PB) include compliance with medical recommendations, avoiding colds, knowledge of emergency numbers, regular medical examinations. Positive mental attitude (PMA) is about avoiding too strong emotions, depressing situations, stress and tension. Health practices (HP) include daily habits relating to rest, sleep, recreation, physical activity, maintaining proper body weight, using stimulants, and smoking. Each behavior was rated on a five-point scale. On the basis of the frequency of indicated health behaviors, the general intensity of the particular categories of health behaviors was determined, as well as the level of intensity of individual categories. The general intensity of these behaviors was the sum of points from the individual categories, within the range from 24 to 120 points. A higher result denotes a greater intensity of declared behaviors.
Verified and modified according to:
· Juczyński Z. Measurement tools in health promotion and psychology. Measure tools in promotion and psychology of health. Warszawa: Laboratory of Psychological Tests of the Polish Psychological Association, Polska, 2012.
· Raykov, T. (1997a). Estimation of Composite Reliability for Congeneric Measures. Applied Psychological Measurement, 21,173-184.
· Raykov, T. (1997b). Scale reliability, Cronbach’s coefficient alpha, and violations of essential tau-equivalence with fixed congeneric components. Multivariate Behavioral Research, 32, 329.353.
· Ciżkowicz B. (2018). McDonald's omega as an alternative to Cronbach's in estimating test reliability. Polskie Forum Psychologiczne, 2018, 23(2), 311-329.
Below we present the estimated reliability of our results of the intensity of health behaviors (Intensity of Health Behaviors)
1. Reliability Analysis:correct eating habits (CEH)
ScaleReliabilityStatistics | ||||
mean | sd | McDonald's ω | Cronbach's α | |
scale | 3.432 | 0.339 | 0.787 | 0.778 |
Note. Of the observations, 437 were used, 4 were excluded listwise, and 441 were provided. |
ItemStatistics
ItemReliabilityStatistics | ||||||||||
Ifitemdropped | ||||||||||
mean | sd | McDonald's ω | Cronbach's α | |||||||
IZZ_1_ | 3.860 | 0.937 | 0.770 | 0.758 | ||||||
IZZ_5_ | 3.114 | 1.099 | 0.738 | 0.725 | ||||||
IZZ_9_ | 3.755 | 1.005 | 0.738 | 0.727 | ||||||
IZZ_13_ | 3.190 | 1.228 | 0.736 | 0.722 | ||||||
IZZ_17_ | 3.103 | 1.193 | 0.750 | 0.737 | ||||||
IZZ_21_ | 3.568 | 1.139 | 0.797 | 0.795 | ||||||
2. Reliability Analysispreventive behaviors(PB)
ScaleReliabilityStatistics | |||||||||
mean | sd | McDonald's ω | Cronbach's α | ||||||
scale | 3.293 | 0.490 | 0.660 | 0.641 | |||||
Note. Of the observations, 435 were used, 6 were excluded listwise, and 441 were provided. | |||||||||
ItemStatistics
ItemReliabilityStatistics | |||||||||
Ifitemdropped | |||||||||
mean | sd | McDonald's ω | Cronbach's α | ||||||
IZZ_2_ | 3.915 | 0.995 | 0.662 | 0.640 | |||||
IZZ_6_ | 2.720 | 1.705 | 0.644 | 0.638 | |||||
IZZ_10_ | 3.743 | 1.129 | 0.575 | 0.558 | |||||
IZZ_14_ | 3.138 | 1.300 | 0.607 | 0.582 | |||||
IZZ_18_ | 2.805 | 1.237 | 0.618 | 0.587 | |||||
IZZ_22_ | 3.439 | 1.187 | 0.608 | 0.580 | |||||
3. Reliability Analysispositive mental attitude (PMA)
ScaleReliabilityStatistics | |||||||||
mean | sd | McDonald's ω | Cronbach's α | ||||||
scale | 3.625 | 0.362 | 0.723 | 0.712 | |||||
Note. Of the observations, 436 were used, 5 were excluded listwise, and 441 were provided. | |||||||||
ItemStatistics
ItemReliabilityStatistics | |||||||||
Ifitemdropped | |||||||||
mean | sd | McDonald's ω | Cronbach's α | ||||||
IZZ_3_ | 3.372 | 1.074 | 0.732 | 0.724 | |||||
IZZ_7_ | 3.640 | 1.079 | 0.638 | 0.624 | |||||
IZZ_11_ | 3.287 | 1.095 | 0.660 | 0.644 | |||||
IZZ_15_ | 4.069 | 1.019 | 0.723 | 0.710 | |||||
IZZ_19_ | 3.323 | 1.091 | 0.658 | 0.644 | |||||
IZZ_23_ | 4.057 | 0.927 | 0.692 | 0.675 | |||||
4. Reliability Analysishealth practices (HP)
ScaleReliabilityStatistics | |||||||||
mean | sd | McDonald's ω | Cronbach's α | ||||||
scale | 3.352 | 0.657 | 0.633 | 0.605 | |||||
Note. Of the observations, 429 were used, 12 were excluded listwise, and 441 were provided. | |||||||||
ItemStatistics
ItemReliabilityStatistics | |||||||||
Ifitemdropped | |||||||||
mean | sd | McDonald's ω | Cronbach's α | ||||||
IZZ_4_ | 3.224 | 1.035 | 0.521 | 0.500 | |||||
IZZ_8_ | 3.203 | 1.181 | 0.562 | 0.516 | |||||
IZZ_12_ | 3.706 | 1.130 | 0.624 | 0.587 | |||||
IZZ_16_ | 3.277 | 1.098 | 0.548 | 0.524 | |||||
IZZ_20_ | 4.350 | 1.182 | 0.659 | 0.639 | |||||
IZZ_24_ | 2.354 | 1.128 | 0.622 | 0.583 | |||||
Reliability Analysis HBI (point score)
ScaleReliabilityStatistics | |||||||||
mean | sd | McDonald's ω | Cronbach's α | ||||||
scale | 3.423 | 0.466 | 0.875 | 0.868 | |||||
Note. Of the observations, 425 were used, 16 were excluded listwise, and 441 were provided. | |||||||||
ItemStatistics
ItemReliabilityStatistics | |||||||||
Ifitemdropped | |||||||||
mean | sd | McDonald's ω | Cronbach's α | ||||||
IZZ_1 | 3.864 | 0.942 | 0.871 | 0.864 | |||||
IZZ_2 | 3.920 | 1.002 | 0.872 | 0.865 | |||||
IZZ_3 | 3.360 | 1.084 | 0.872 | 0.865 | |||||
IZZ_4 | 3.221 | 1.038 | 0.870 | 0.863 | |||||
IZZ_5 | 3.108 | 1.102 | 0.868 | 0.861 | |||||
IZZ_6 | 2.727 | 1.714 | 0.874 | 0.872 | |||||
IZZ_7 | 3.649 | 1.080 | 0.867 | 0.860 | |||||
IZZ_8 | 3.202 | 1.186 | 0.871 | 0.865 | |||||
IZZ_9 | 3.748 | 1.012 | 0.864 | 0.858 | |||||
IZZ_10 | 3.734 | 1.132 | 0.868 | 0.861 | |||||
IZZ_11 | 3.282 | 1.097 | 0.865 | 0.858 | |||||
IZZ_12 | 3.713 | 1.132 | 0.868 | 0.862 | |||||
IZZ_13 | 3.181 | 1.232 | 0.866 | 0.859 | |||||
IZZ_14 | 3.122 | 1.294 | 0.869 | 0.862 | |||||
IZZ_15 | 4.071 | 1.023 | 0.875 | 0.868 | |||||
IZZ_16 | 3.273 | 1.101 | 0.870 | 0.864 | |||||
IZZ_17 | 3.096 | 1.191 | 0.868 | 0.861 | |||||
IZZ_18 | 2.805 | 1.239 | 0.869 | 0.862 | |||||
IZZ_19 | 3.318 | 1.095 | 0.868 | 0.861 | |||||
IZZ_20 | 4.351 | 1.184 | 0.875 | 0.869 | |||||
IZZ_21 | 3.562 | 1.142 | 0.873 | 0.866 | |||||
IZZ_22 | 3.438 | 1.192 | 0.871 | 0.863 | |||||
IZZ_23 | 4.052 | 0.933 | 0.872 | 0.865 | |||||
IZZ_24 | 2.348 | 1.125 | 0.875 | 0.868 | |||||
The interaction of factors produces an additional effect (Table 8). Sex, age, education, material situation, frequency and history of training had an impact on the intensity of health behaviors of CS and MA athletes. The two-factor interaction between sport discipline and sex was modified by age factor (CEH, PB, PMA) and the interaction between sport and age was modified by sex (PMA). The two-factor interaction between sports discipline and frequency of exercises was modified by sporting experience (CEH, HP) and sporting experience and sex (CEH, PB, HP).
Three-factor interactions: between sport discipline, age and education was modified by sex (CEH); between sport, age and sex was modified by financial situation (CEH, PB); between sport, age and financial situation was modified by sex (HP); interaction between sport, frequency and seniority is was modified by sex (CEH, PB, PMA). We only found an interaction between sport discipline, weekly frequency of training and sporting experience (PMA).
Sex was the factor which most often modified the analyzed interactions (two- and three-factor interactions).
Table 8. Statistically significant interactions between the selected health behavior categories (confirmatory factor analysis)
Combinations of factors | Interactions | Categories of health behaviors | |||||||
CEH | PB | PMA | HP | ||||||
F | p | F | p | F | p | F | P | ||
sport*age* sex | sport*sex | 4.30 | 0.0020 | 2.73 | 0.0289 | 3.31 | 0.0109 | ||
sport*age | 1.83 | 0.0411 | |||||||
sport*age*sex*education | sport*age*education | 2.52 | 0.0409 | ||||||
sport*age* sex*financial situation | sport*age* sex | 7.13 | 0.0079 | 5.24 | 0.0227 | ||||
sport*age* financial situation | 2.11 | 0.0233 | |||||||
sport* weekly frequency*sporting experience | sport*weekly frequency* sporting experience | 1.66 | 0.0368 | ||||||
sport* weekly frequency | 3.77 | 0.0024 | 3.06 | 0.0101 | |||||
sport*sporting experience*weekly frequency*sex | sport* weekly frequency | 8.22 | 0.0044 | 5.03 | 0.0256 | 4.75 | 0.0300 | ||
sport*sporting experience*weekly frequency | 1.96 | 0.0314 | 1.87 | 0.0425 | 2.75 | 0.0020 |
We hope that above explanation and corrections done have met your expectations.
Best regards
Katarzyna Kotarska
